# Analysis of the Dynamic Evolution Game of Government, Enterprise and the Public to Control Industrial Pollution

Na Yu * and Meilin Lu

College of Economics and Management, Hefei University, Hefei 230601, China
* Correspondence: yn2016@hfuu.edu.cn

**Abstract:** This paper proposes a two-party evolutionary game model of government and enterprise to solve the dilemma of industrial pollution control and explore the mode of government and enterprise collaborative governance. The local equilibrium points of the game model in four cases are calculated and analyzed, and the results show that government power alone cannot help enterprises achieve an ideal level of pollution reduction, and it is necessary to introduce public power for supervision. Based on the above, a tripartite evolutionary game model comprising the government, the public, and the enterprise is proposed. When the costs and benefits of the tripartite game players meet certain conditions, the system will evolve to a state of equilibrium (0,1,1). Following the current situation of economic development in China, the parameters of the two-party and tripartite evolutionary game are assigned, and the operating path and system's evolution trajectory of the two-party and tripartite industrial pollution control are simulated by Matlab R2016a software. It is indicated that whether the government participates in supervision or not, an enterprise will actively control pollution under strong public supervision, which can provide feasible suggestions for the selection of industrial pollution control policies.

**Keywords:** industrial pollution; pollution supervision; evolution game; public participation; system dynamics

## 1. Introduction

With the continuous deepening development of industrialization in China, industrial production needs a lot of resources and energy [1,2], along with the production and emission of large amounts of wastewater, waste gas, waste residue, especially strong toxic, harmful industrial pollutants, etc. [3,4], which will cause serious damage to the ecological environment [5]. Although the Chinese government pays great attention to sustainable development, industry pollution governance has not attracted enough attention. Simultaneously, the efficiency of the current industrial pollution supervision in China is far lower than that of the Western developed countries and regions.

Industrial pollution treatment is a very complicated project, which involves the interest relationship between the government, enterprises, and the public [6]. At present, it is a very urgent matter to improve the supervision and governance system of industrial pollution. However, it is unclear what standards and measures the Chinese government uses in terms of supervision and governance. How to make full use of the relevant national policies to achieve effective and severe supervision of industrial pollution has become an urgent problem to be resolved on the background of the lagging development of industrial pollution supervision. Therefore, this study established an evolutionary game model that studied the interests among different entities and provided a reference for scientific decision-making by the government.

Many scholars have also studied the strategy of pollution control from micro, meso, and macro perspectives. For example, Van der Kamp proposed a social cost-benefit analysis model to measure the optimal level of pollution, which has been widely used in regulatory

impact assessments [7]. Zhao et al. established a plant-level aggregation method to calculate the spatial distribution characteristics of water pollution and assess the pollution status of newly added industrial projects using discrete event simulation [8]. Xu et al. constructed an integrated water and waste load allocation model to balance the relationship between economic growth, environmental protection, and resource utilization. The practicality and effectiveness of the model were verified by taking industrial parks as an example [9]. Xie et al. established an inexact two-stage stochastic downside risk-aversion model to analyze the impact of risk avoidance attitudes on pollutant emissions in different industries. The results indicated that the total control of pollutant emissions and expected benefits could serve as effective measures for regional industrial structure adjustment [10]. Yano and Sakawa introduced the concepts of the entire M-Pareto optimality and the decision powers to study the pollution control dilemma in Osaka City in Japan [11]. Ebiefung and Udo presented a controlling industrial pollution model based on the Leontief input–output production and pollution coefficients, which was used in multiple developed economies such as the United States [12]. Yang et al. analyzed the impact of digitalization on industrial pollution using data from Chinese-listed companies from 2010 to 2020, including pollutant data and financial data. The results showed that digital development could promote a decrease in pollutant emissions, mainly derived from structural and technical factors [13]. Lu et al. combined environmental and economic characteristic data from enterprises, industrial parks, etc., to construct a panel data model and empirically analyze the impact of industrial parks on environmental pollution. The results indicated that the expansion of production scale and an increase in pollution-intensive industries were the main factors causing more severe regional pollution [14]. Li et al. conducted research on the microdata of industrial enterprises, using fixed effects models and quantile regression models to explore the synergistic benefits of air pollution and greenhouse gas emissions reduction. The analysis demonstrated that enterprise emission reduction behavior and government emission reduction legislation had positive effects on pollution control [15].

Some scholars studied controlling industrial pollution based on a meso perspective. They adopted data envelopment analysis (DEA) and stochastic frontier analysis (SFA) to measure industrial pollution control efficiency [16–19]. Some scholars used the system dynamics (SD) model to connect the feedback mechanisms of industrial sectors [20]. Some scholars used the evolutionary game model to analyze the relationship between local governments and polluting enterprises; it was found that the game order and initial endowment between local governments and polluting enterprises determine whether their relationship in environmental governance is cooperative or collusive [21]. Guo et al. constructed a technology finance index at the urban level, combining data from Chinese industrial enterprises and other city-level data. The study indicated that financial technology can significantly reduce pollution emissions [22]. Song et al. empirically analyzed the relationship between industrial agglomeration and air pollution using panel data from 277 prefecture-level cities. The results showed that industrial agglomeration not only had a direct impact on air pollution but also had an indirect impact on air pollution through technological innovation [23]. Chen et al. explored the spatiotemporal evolution characteristics of pollution and carbon reduction at the urban level and found that national industrial ecological parks have significant spatial spillover effects [24]. Hao et al. collected balanced panel data from 164 cities from 2003 to 2013 in China and empirically analyzed the relationship between industrial agglomeration and air pollution. The results denoted that industrial agglomeration could affect environmental regulation, technological progress, and industrial structure upgrading, which can reduce the level of air pollution in urban agglomerations [25]. Wu et al. studied the impact of digital economy development on air pollution in 274 cities in China, and the results showed that the digital economy could effectively promote industrial agglomeration in various regions. The elasticity of pollution reduction in the central and western regions was higher than that in the eastern regions [26]. Lan et al. explored the impact of green finance on industrial pollution emissions using

data from 30 provinces from 2001 to 2020, and the results showed a regional heterogeneity relationship between green finance and pollutant emissions [27].

Some scholars also measured the effectiveness of industrial pollution control from a macro perspective. For example, Cheng and Xu analyzed the urban industrial pollution emissions and environmental quality from fiscal policy, and they found that the vertical reform of environmental administrations could effectively improve pollution emissions, which had a more significant effect in western-region cities and small and medium-sized cities [28]. Bai et al. examined the impact of large-scale and low-priced transfers of industrial land by local governments on pollution emissions from Chinese industrial enterprises, and they found that there was a positive correlation between large-scale, low-cost industrial land conversion and industrial pollutant emissions. Further analysis suggested that a misallocation of land resources and illegal land laws exacerbated industrial pollution [29]. Xu et al. explored the impact of sub-provincial taxation on industrial pollution. The results showed that the increase in value-added tax allocation exacerbated environmental pollution, and the phenomenon in the central and western regions was more prominent [30]. Hao et al. studied the heavy metal pollution caused by industrial transfer, mainly reflected in the western, peripheral, and southern regions [31]. Du and Li studied environmental target constraint policies. The implementation of environmental target constraint policies effectively reduced the emissions of $SO_2$ and other pollutants from polluting enterprises, which helped to optimize the energy structure [32]. Shen et al. studied the spatiotemporal distribution pattern of industrial wastewater discharge in North China using exploratory spatial data analysis (ESDA) and spatial econometric models; the results showed that the industrial structure promoted wastewater discharge [33].

Recently, industrial pollution control has attracted increasing attention from governments, society, and scholars. Nowadays, the industrial environment of various countries has been greatly improved, but many problems have not been resolved. Many enterprises have reached the pollution standard but still have adverse effects on the environment and public life. Solving the problem of industrial pollution is still an important part of the sustainable development of countries [4,34]. Although some domestic and foreign scholars have conducted a lot of research on the prevention and governance of industrial pollution on the background of the strategy of carbon dioxide emissions and carbon neutrality [35], most of them focus on the formulation of national policies and when at the level of public participation, there are relatively few studies on how to manage industrial pollution efficiently. Based on the results of the previous research, this paper intends to analyze the evolutionary game of government and enterprise industrial pollution in regulatory supervision, use system dynamics to conduct empirical research on evolutionary decisions with system dynamics, as well as put forward constructive policy suggestions. The flowchart is shown in Figure 1.

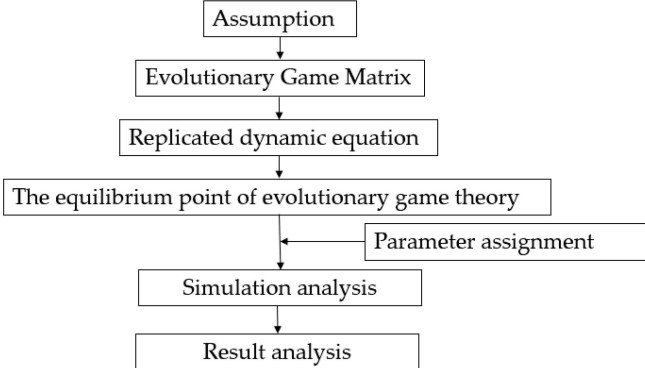

**Figure 1.** The flowchart of the dynamic evolutionary game.

## 2. Materials and Methods

Genetic ecologists Fisher, Hamilton, T. Five, and others have studied the cooperative and conflicting behavior of animals and plants and found that game theory can explain most of the evolutionary outcomes, which was further supported by the absence of any rational assumptions. They believe that an evolutionary game is a dynamic model that exists between cooperative and conflicting behaviors among species. This model can reflect the laws of interactions and evolution between species, providing a new method for future research on evolutionary phenomena and mechanisms. In 1973, Smith and Price first proposed the evolutionary stable strategy (ESS), and evolutionary game theory was officially born at this time.

For the traditional game theory, it is assumed that the participants are sufficient, reasonable, and completely rational, but both of these assumptions are difficult to satisfy in the real economy. In any type of cooperative, competitive relationship, the behavior of participants will differ greatly under incomplete information conditions due to economic factors and an understanding of the game problem.

From the perspective of an evolutionary game [36], the game situation depends on the behavior of participants [37]. With the changes in the situation, individuals maintain the trend of interaction between groups and show dynamic changes [38]. In the game of government and enterprises, companies will inevitably pay more attention to their income to achieve their interests. Therefore, each interested party can continue to make choices and amendments and finally seek the best-balanced solution only in the continuous game.

In the process of evolution of the two main subjects of the government and enterprises, the government plays a macro-regulating function in terms of economic aspects but also plays the regulatory function of local governments and enterprises in the ecological environment. Each local government has extremely important responsibilities for the environmental supervision of the region. From an economic point of view, it is unrealistic to rely on public power to solve the problem of ecological environmental protection. Moreover, the government's supervision of the ecological environment can effectively reduce the probability of market failure, and the government has played a role in the supervision and control of ecological protection. However, when the local governments pursue local economic development, they will not be in a place to implement the central policy documents, so the governments will not supervise enterprises, especially when enterprises have reached pollution standards, and the government implements a lower motivation for supervision. Therefore, in the process of governance of enterprises, the government employs two strategies: active supervision strategy and negative supervision strategy.

From the perspective of the enterprise, the goals of companies are not only to seek the maximum economic benefits but also to pursue both economic benefits and social benefits. At the same time, public opinion has a direct impact on social benefits, so some companies pay some costs for obtaining good social benefits. For an enterprise, if its economic income has been guaranteed, it introduces more advanced pollution control and sewage facilities in accordance with the indicators of the government, and minimizes the damage caused by pollution to the life and environment of the public. However, the government will also give enterprises a certain amount of subsidies. Therefore, enterprises also have two strategies, namely active pollution control strategies and negative pollution control strategies.

### 2.1. The Two-Party Game Model Assumption

In the case of the participation of both the government and the enterprise, the following assumptions are first made:

**Hypothesis 1 (H1):** *Participating subject. In the evolutionary game model, there are two participants, one is the government departments, and the other is an enterprise.*

**Hypothesis 2 (H2):** *Cooperative strategy. This is assuming that the government has the right to control the enterprises, to improve the indicators of the enterprises' emissions to the level of*

*environmental protection, and the pollution emitted by the enterprises will not affect the lives of the public and the environment. It is also possible that the government only pays attention to one-sided economic development and ignores the protection of the ecological environment and public interests [39]. Enterprises can choose to actively control pollution, which not only meets environmental protection standards but also minimizes pollution. Enterprises can also choose not to actively control pollution, that is, passive pollution control; they can only meet the minimum standards of pollution control, ignoring the actual needs of the public. Therefore, both sides of the game have two strategies: the government's strategy is active regulation or negative regulation, and the enterprise's strategy is active pollution control or negative pollution control. The strategic space of this game is active regulation, positive pollution control; active regulation, negative pollution control; negative regulation, active pollution control; and negative regulation, negative pollution control.*

**Hypothesis 3 (H3):** *Government benefits and cost. Both parties of the game are rational economists with their interests, and all the benefits, costs, and expenses in the model can be quantified [40]. The government's choice of active regulation and negative regulation strategy mainly depends on the government's profit expectations [41,42]. The process of enterprise pollution control involves basic enterprise revenue, regulatory costs, and incentives. The difference between the government's income and regulatory costs during passive regulation can be expressed by $R_G$, and the additional cost of government active regulation can be expressed by $C_G$. If the enterprise is passive in pollution control, the government will bear the corresponding treatment costs, assuming that it is expressed by $D$. Government incentive measures mean that if enterprises actively control pollution, the government will give certain subsidies [43]. The subsidy will be issued according to the "regression" mechanism; that is, the subsidy amount in the process of industrial pollution supervision will decrease with time. Assuming that the enterprise chooses to actively control pollution, then the subsidy issued by the government to the enterprise is $f(n)$, where this is a decreasing function, $0 < f(n) < 1$. The full maximum subsidy of the government can be expressed as $B$, where the subsidy uses the "regression" mechanism to enable enterprises to obtain subsidies in batches. During this period, enterprises must actively control pollution to obtain subsidies at each stage.*

**Hypothesis 4 (H4):** *Enterprise benefits and cost. The enterprise has two strategic options: active pollution control and negative pollution control. If the enterprise chooses to actively control pollution, the difference between the normal income of the enterprise and the cost of pollution control is $R_E$, the additional cost of the enterprise is $C_{E1}$, and the subsidy received by the enterprise is $f(n)B$. If the enterprise chooses passive pollution control, the penalty received is recorded as $J$.*

The variables involved in the industrial pollution regulatory game between the government and enterprises are shown in Table 1.

**Table 1.** Variables involved in the evolutionary game of industrial pollution regulation between the government and enterprises.

| Variable | Variable Meaning |
|---|---|
| $R_G$ | The difference between the normal income and the cost of regulation when the government is passively regulated |
| $C_G$ | The additional cost to the government of active regulation |
| $D$ | Governance costs borne by the government |
| $J$ | The punishment received by enterprises for choosing passive pollution control |
| $f(n)$ | A function in which the dependent variable decreases as the independent variable increases |
| $B$ | The full maximum subsidy provided by the government to enterprises |
| $R_E$ | The difference between the normal income and the cost of pollution control obtained by the enterprise in passive pollution control |
| $C_{E1}$ | The additional costs incurred by enterprises in actively controlling pollution |

### 2.2. The Two-Party Game Model

To better explain the supervision of industrial pollution and enterprise pollution control for the government, it is assumed that both the government and the enterprise act according to a certain probability and $x$ is used to represent the probability of active and negative supervision by the government, where $x = 0$ means that the government adopts a completely passive supervision strategy, and $x = 1$ means the government adopts a completely active supervision strategy. The closer $x$ is to 1, the higher the activity of government supervision. $y$ represents the probability of the active and passive pollution control of the enterprise, where $y = 0$ indicates that the enterprise adopts a completely negative pollution control strategy and $y = 1$ indicates that the enterprise adopts a completely active pollution control strategy. The closer $y$ is to 1, the higher the pollution control activity of the enterprise. At the same time, the probability range of government supervision and enterprise pollution control is: $0 < x < 1, 0 < y < 1$. Table 2 shows the revenue matrix of the government–enterprise industrial pollution regulation game model.

**Table 2.** Government and enterprise industrial pollution supervision game model's benefits matrix.

| Government | Enterprise | |
|---|---|---|
| | Positive Pollution $y$ | Negative Pollution $1-y$ |
| Actively supervise $x$ | $(R_G - C_G - f(n)B, R_E + f(n)B - C_{E1})$ | $(R_G + J - D - C_G, R_E - J)$ |
| Negative supervision $1 - x$ | $(R_G, R_E - C_{E1})$ | $(R_G - D, R_E)$ |

### 2.3. The Two-Party Game Model Analysis

#### 2.3.1. The Government Game Model Solution

According to the game model assumptions of the governments and enterprises and the game matrix, the average income of the government and the enterprise can be obtained. The government adopts the expected income of different strategies, and its average income and the dynamic equation of the execution supervision are expressed as follows [44]:

The expected income of the government under active supervision is expressed as follows:

$$U_{G1} = y \cdot (R_G - C_G - f(n) \cdot B) + (1 - y)(R_G + J - C_G - D) \tag{1}$$

The expected income of the government during the negative supervision is given as follows:

$$U_{G2} = y \cdot R_G + (1 - y)(R_G - D) \tag{2}$$

The average income of the government is expressed as follows:

$$U_G = x \cdot U_{G1} + (1 - x)U_{G2} \tag{3}$$

The dynamic equation when the government executes supervision is expressed as follows:

$$F(x) = \frac{dx}{dt} = x \cdot (U_{G1} - U_G) = x(1 - x)(U_{G1} - U_{G2}) = x(1 - x)[J - C_G - y(J + f(n)B)] \tag{4}$$

The direction to $F(x)$ is as follows:

$$F'(x) = (1 - 2x)[J - C_G - y(f(n)B + J)] \tag{5}$$

Let $F(x) = 0$, $y^* = \frac{J - C_G}{f(n)B + J}$, and evolve the stable analysis according to the range of the $y^*$ value.

(1) When $F(x) = 0$, $y^* = \frac{J - C_G}{f(n)B + J}$, no matter what the value $x$ takes, and $F(x) \equiv 0$, and $F'(x) \equiv 0$, indicating that the government is in a stable state at this time.

(2) If $\frac{J - C_G}{f(n)B + J} < 0$, then $J < C_G$; that is, the social income obtained by the government is less than the government's supervision cost. At this time, $F'(0) < 0$ and $F'(1) > 0$, and

$x = 0$ is the only stable evolution strategy; that is, the government's evolutionary stability strategy is a negative regulatory strategy.

(3) When $0 < \frac{J-C_G}{f(n)B+J} < 1$, $J > C_G$, and $f(n)B > -C_G$; that is, the social income obtained by the government is greater than the government's regulatory cost:

① When $y > \frac{J-C_G}{f(n)B+J}$ and $F(x) = 0$, two stable points are $x = 0$ and $x = 1$. At this time, there is $F'(0) < 0$ and $F'(1) > 0$, and the only evolutionary stability strategy is $x = 0$. It shows that when the company's active pollution rate is greater than $\frac{J-C_G}{f(n)B+J}$, regardless of the government's initial strategy, the government's final evolutionary stability strategy is negative supervision.

② When $y < \frac{J-C_G}{f(n)B+J}$ and $F(x) = 0$, two stability points are $x = 0$ and $x = 1$. At this time, there is $F'(0) > 0$ and $F'(1) < 0$, and the only evolution stability strategy is $x = 1$. It shows that when an enterprise's active pollution is less than $\frac{J-C_G}{f(n)B+J}$, the government's final evolutionary stability strategy is active supervision.

In summary, when the probability of pollution control is equal $\frac{J-C_G}{f(n)B+J}$, the government is in a stable state regardless of the value $x$. When the probability of pollution control is greater than $\frac{J-C_G}{f(n)B+J}$, the government's strategy of evolutionary stability is negative supervision. When the probability of pollution control is less than $\frac{J-C_G}{f(n)B+J}$, the government's strategy of evolutionary stability is active supervision.

2.3.2. The Enterprise Game Model Solution

The expected income of different strategies can be obtained, and dynamic equations for enterprise pollution control are copied according to the government's and enterprise's game model assumptions and the game matrix. The expected return of an enterprise's active pollution control is given as follows:

$$U_{E1} = x \cdot (R_E + f(n) \cdot B - C_{E1}) + (1 - x)(R_E - C_{E1}) \tag{6}$$

The expected income from the negative pollution control of enterprises is:

$$U_{E2} = x \cdot (R_E - J) + (1 - x)R_E \tag{7}$$

The average income of enterprise pollution control is:

$$U_E = y \cdot U_{E1} + (1 - y)U_{E2} \tag{8}$$

The dynamic equation of positive pollution control is:

$$F(y) = \frac{dy}{dt} = y(U_{E1} - U_E) = y(1 - y)(xJ - C_{E1} + xf(n)B) \tag{9}$$

With this direction to $F(y)$, we can obtain the following result:

$$F'(y) = (1 - 2y)(xJ - C_{E1} + xf(n)B) \tag{10}$$

Let $F(y) = 0$, then we can obtain $y = 0$, $y = 1$, and $x^* = \frac{C_{E1}}{f(n)B+J}$. We can analyze the stability according to the different value intervals of $x^*$.

(1) When $x^* = \frac{C_{E1}}{f(n)B+J}$, no matter what the initial value of $y$, $F(y) \equiv 0$, and $F'(y) \equiv 0$; this shows that the enterprise is always in a stable state.

(2) When $\frac{C_{E1}}{f(n)B+J} > 1$, there is $x < \frac{C_{E1}}{f(n)B+J}$. If $F(y) = 0$, we can obtain $y = 0$, $y = 1$, and then obtain $F'(0) < 0$, $F'(1) > 0$. The only evolutionary stability strategy is $y = 0$. It shows that when the government supervision probability is less than $\frac{C_{E1}}{f(n)B+J}$, negative pollution has become a strategy of evolutionary stability in enterprises.

(3) When $0 < \frac{C_{E1}}{f(n)B+J} < 1$

① When $x < \frac{C_{E1}}{f(n)B+J}$ and $F(y) = 0$, we can obtain two stable points of $y = 0$ and $y = 1$. At this time, there is $F'(0) < 0$ and $F'(1) > 0$. The only evolutionary stability strategy is $y = 0$. It shows that when the government supervision probability is less than $\frac{C_{E1}}{f(n)B+J}$, the evolutionary stability strategy selected by the enterprise is negative control pollution.

② When $x > \frac{C_{E1}}{f(n)B+J}$ and $F(y) = 0$, we can obtain two stable points of $y = 0$ and $y = 1$. At this time, there is $F'(0) > 0$ and $F'(1) < 0$. The only evolutionary stability strategy is $y = 1$. It shows that when the government supervision probability is more than $\frac{C_{E1}}{f(n)B+J}$, the enterprise chooses to actively control pollution to evolve the stability strategy.

In summary, when the probability of government supervision is equal to $\frac{C_{E1}}{f(n)B+J}$, regardless of the initial value of $y$, the enterprise is always in a state of stability. When the government supervision probability is greater than $\frac{C_{E1}}{f(n)B+J}$, the active pollution control is the strategy of evolutionary stability. When the government supervision probability is less than $\frac{C_{E1}}{f(n)B+J}$, negative pollution control is the strategy of evolutionary stability.

### 2.3.3. The Two-Party Game Model Solution

The two parts are jointly organized to obtain the dynamic equation of the government and enterprises, and they are as follows:

$$F(x) = x(1-x)[J - C_G - y(f(n)B + J)] \tag{11}$$

$$F(y) = y(1-y)(xJ - C_{E1} + xf(n)B) \tag{12}$$

$$F'(x) = (1-2x)[J - C_G - y(f(n)B + J)] \tag{13}$$

$$F'(y) = (1-2y)(xJ - C_{E1} + xf(n)B) \tag{14}$$

Let $F(x) = F(y) = 0$. Then, we can obtain five stable points; that is, $(0,0)$, $(0,1)$, $(1,0)$, $(1,1)$, and $(x^*, y^*)$. The derivation of dynamic equations leads us to obtain the following:

$$\frac{dF(x)}{dx} = (1-2x)[J - C_G - y(f(n)B + J)] \tag{15}$$

$$\frac{dF(x)}{dy} = x(1-x)(f(n)B + J) \tag{16}$$

$$\frac{dF(y)}{dx} = y(1-y)(\text{f(n)}B + J) \tag{17}$$

$$\frac{dF(y)}{dy} = (1-2y)(xJ - C_{E1} + xf(n)B) \tag{18}$$

It can be seen from the theory of evolution game that the stability of Nash equilibrium can be obtained through the Jacobian matrix. The Jacobian matrix is expressed as follows:

$$J = \begin{bmatrix} (1-2x)[J - C_G - y(J + f(n)B)] & -x(1-x)(f(n)B + J) \\ y(1-y)(f(n)B + J) & (1-2y)[xJ - C_{E1} + xf(n)B] \end{bmatrix} \tag{19}$$

Bring $A(0,0)$ into the Jacobian matrix and obtain:

$$J_A = \begin{bmatrix} J - C_G & 0 \\ 0 & -C_{E1} \end{bmatrix} \tag{20}$$

In the same way, bring the other four points into the Jacobian matrix:

$$J_B = \begin{bmatrix} -f(n)B - C_G & 0 \\ 0 & C_{E1} \end{bmatrix} \tag{21}$$

$$J_C = \begin{bmatrix} C_G - J & 0 \\ 0 & J - C_{E1} + f(n)B \end{bmatrix} \quad (22)$$

$$J_D = \begin{bmatrix} C_G + f(n)B & 0 \\ 0 & C_{E1} - J - f(n)B \end{bmatrix} \quad (23)$$

$$J_E = \begin{bmatrix} 0 & b_{12} \\ b_{21} & 0 \end{bmatrix} \quad (24)$$

Among which:

$$b_{12} = -C_{E1}\left(1 - \frac{C_{E1}}{f(n)B + J}\right) \quad (25)$$

$$b_{21} = (J - C_G)\left(1 - \frac{J - C_G}{f(n)B + J}\right) \quad (26)$$

$$DetJ = \frac{dF(x)}{dx} \cdot \frac{dF(y)}{dy} - \frac{dF(x)}{dy} \cdot \frac{dF(y)}{dx} = a_{11}a_{22} - a_{12}a_{21} \quad (27)$$

$$TrJ = \frac{dF(x)}{dx} + \frac{dF(y)}{dy} = a_{11} + a_{22} \quad (28)$$

*2.4. The Tripartite Game Model Assumption*

In the context of carbon dioxide emissions and carbon neutrality, the public has put forward new and higher-level requirements for sustainable development and environmental protection. Therefore, regulating industrial pollution has become an urgent task. In recent years, with the downward shift of the focus of social governance, various regions have organically combined grassroots conflict mediation, resulting in many practical innovations that can improve the level of public management. Meantime, it has also provided some reference areas for the development of the industrial pollution supervision industry. By various means, we can improve the level of public participation in supervision, fully mobilize the enthusiasm for grassroots supervision, and build a regulatory community.

However, there are some problems for the public in pollution regulation. For example, with the continuous improvement of the industrial pollution supervision system in China, the current laws, regulations, and rules can neither meet the practical needs of industrial pollution supervision nor can they meet the practical needs of public participation in supervision. In the reality of public participation in industrial pollution regulation, their initiative and enthusiasm to participate in industrial pollution regulation are greatly reduced due to their weak subjective will and unclear boundaries of rights and responsibilities.

We selected the government, enterprises, and the public as the main bodies of interest and built a tripartite evolution game model. The subject of the tripartite faces different strategic choices. If the government, enterprises, and the public form a virtuous circle in the process of evolution of the tripartite, we must seek a balanced mechanism that can avoid the negative impact of the enterprise on the social environment. The following assumptions of the game model of the government, enterprises, and the public are as follows:

**Hypothesis 5 (H5):** *Government strategy. The government is facing two strategies: active supervision and negative supervision. If the government takes active supervision, it is necessary to pay additional costs $C_G$. The difference between the normal income and regulatory cost of the government during negative supervision is $R_G$. When the enterprise chooses negative pollution control, the government's governance cost is D. The government gives the subsidies for active pollution control enterprises to be $f(n)B$, and the government's subsidy to the public is M.*

**Hypothesis 6 (H6):** *Enterprise strategy. The enterprise faces two strategies: active pollution control and negative pollution control. Assume that the difference between the normal income and the cost of pollution control when the enterprise implements negative pollution is $R_E$. When the enterprise spends additional costs when actively treating pollution, it is recorded as $C_{E1}$. When*

*choosing negative pollution control, the punishment by the enterprise is represented by J. The loss of non-active pollution control to enterprises is recorded as N.*

**Hypothesis 7 (H7):** *Public strategy. The public faces two strategies: participating in supervision or not participating in supervision. Under the supervision of the public's participation, the cost of its supervision is L, and when the public participates in supervision, the government's subsidy is M. If it is not active in pollution control, the harm to the public will be recorded as I.*

**Hypothesis 8 (H8):** *Strategy probability. The proportion of the government's choice of active supervision is β, the proportion of the enterprise's selection of active pollution is β, and the proportion of the public choosing to participate in the supervision is γ. Among them, $0 \leq \alpha \leq 1$, $0 \leq \beta \leq 1$, and $0 \leq \gamma \leq 1$.*

The variables involved in the evolution of government, enterprise, and public industrial pollution supervision are shown in Table 3.

**Table 3.** Variables involved in the evolutionary game of government, enterprise, and the public.

| Variable | Variable Meaning |
|---|---|
| $R_G$ | The difference between the normal income and regulatory cost of the government during negative supervision |
| $C_G$ | When implementing active supervision, the government spends additional costs |
| $D$ | Governance costs borne by the government |
| $J$ | The punishment received by the company when choosing negative pollution |
| $f(n)$ | Functions that are constantly decreasing due to the increase in the variables |
| $B$ | The full subsidy of the government to the enterprise |
| $R_E$ | The difference between the normal income and pollution control cost during the time of negative pollution |
| $C_{E1}$ | Enterprise's additional cost during active pollution treatment |
| $L$ | Public supervision cost |
| $M$ | The government's subsidies when participating in supervision |
| $I$ | The adverse effects on the public when the enterprise is not actively treating pollution |
| $N$ | Losses caused by non-active pollution control |

*2.5. The Tripartite Game Model*

Based on the hypothesis of the tripartite evolution game, the income matrix between enterprises, governments, and the public, under the combination of different game strategies, is shown in Table 4.

**Table 4.** Industrial pollution supervision tripartite game model's benefits matrix.

| Government | Enterprise | | | |
|---|---|---|---|---|
| | Positive Pollution Control $\beta$ | | Negative Pollution Control $1-\beta$ | |
| | Public | | | |
| | Participate in Supervision $\gamma$ | Non-Participate in Supervision $1-\gamma$ | Participate in Supervision $\gamma$ | Non-Participate in Supervision $1-\gamma$ |
| Actively supervision $\alpha$ | $\begin{pmatrix} R_G - C_G - f(n)B - M, \\ R_E + f(n)B - C_{E1}, \\ M - L \end{pmatrix}$ | $\begin{pmatrix} R_G - C_G - f(n)B, \\ R_E + f(n)B - C_{E1}, \\ 0 \end{pmatrix}$ | $\begin{pmatrix} R_G + J - C_G - D - M, \\ R_E - J - N, \\ M - L - I \end{pmatrix}$ | $\begin{pmatrix} R_G + J - C_G - D, \\ R_E - J, \\ -I \end{pmatrix}$ |
| Negative supervision $1 - \alpha$ | $\begin{pmatrix} R_G, \\ R_E - C_{E1}, \\ -L \end{pmatrix}$ | $\begin{pmatrix} R_G, \\ R_E - C_{E1}, \\ 0 \end{pmatrix}$ | $\begin{pmatrix} R_G - D, \\ R_E - N, \\ -L - I \end{pmatrix}$ | $\begin{pmatrix} R_G - D, \\ R_E, \\ -I \end{pmatrix}$ |

2.5.1. The Dynamic Copy Equation

The average expected income of the government, enterprises, and the public is expressed in $E_1$, $E_2$, and $E_3$, respectively.

(1) Government game strategy

The average income of the government's choice of actively supervising the game strategy is expressed as follows:

$$
\begin{aligned}
E_{11} &= \beta\gamma(R_G - C_G - f(n)B - M) + \beta(1-\gamma)(R_G - C_G - f(n)B) \\
&+ \gamma(1-\beta)(R_G + J - C_G - D - M) + (1-\beta)(1-\gamma)(R_G + J - C_G - D) \\
&= R_G + J + \beta D - C_G - D - \beta J - \gamma M - \beta f(n)B
\end{aligned} \tag{29}
$$

The average income of the government's choice of negative supervision game strategies is given in:

$$
\begin{aligned}
E_{12} &= \beta\gamma R_G + \beta(1-\gamma)R_G + \gamma(1-\beta)(R_G - D) + (1-\beta)(1-\gamma)(R_G - D) \\
&= R_G - D + \beta D
\end{aligned} \tag{30}
$$

The average expected income of the government is as follows:

$$
\begin{aligned}
E_1 &= \alpha E_{11} + (1-\alpha)E_{12} \\
&= R_G + \beta D + \alpha J - D - \alpha C_G - \alpha\beta J - \alpha\gamma M - \alpha\beta f(n)B
\end{aligned} \tag{31}
$$

The government's replication dynamic differential equation is as follows:

$$
\begin{aligned}
F(\alpha) &= \frac{d\alpha}{dt} = \alpha(E_{11} - E_1) = \alpha(1-\alpha)(E_{11} - E_{12}) \\
&= \alpha(1-\alpha)(J - C_G - \beta J - \gamma M - \beta f(n)B)
\end{aligned} \tag{32}
$$

(2) Enterprise game strategy

The expected income when an enterprise chooses to actively engage in pollution control strategies is as follows:

$$
\begin{aligned}
E_{21} &= \alpha\gamma(R_E + f(n)B - C_{E1}) + \gamma(1-\alpha)(R_E - C_{E1}) \\
&+ \alpha(1-\gamma)(R_E + f(n)B - C_{E1}) + (R_E - C_{E1})(1-\alpha)(1-\gamma) \\
&= \alpha f(n)B + R_E - C_{E1}
\end{aligned} \tag{33}
$$

The expected income when an enterprise chooses a negative pollution strategy is as follows:

$$
\begin{aligned}
E_{22} &= \alpha\gamma(R_E - C_{E2} - N) + (1-\alpha)\gamma(R_E - C_{E2} - N) \\
&+ (R_E - C_{E2})\alpha(1-\gamma) + (1-\alpha)(1-\gamma)(R_E - C_{E2}) \\
&= R_E - \alpha J - \gamma N
\end{aligned} \tag{34}
$$

The average expected income of the enterprise is as follows:

$$
E_2 = \beta E_{21} + (1-\beta)E_{22} = \alpha\beta f(n)B - \beta C_{E1} + R_E - \alpha J + \alpha\beta J - \gamma N + \beta\gamma N \tag{35}
$$

The dynamic differential equation of the enterprise is as follows:

$$
\begin{aligned}
F(\beta) &= \frac{d\beta}{dt} = \beta(E_{21} - E_2) = \beta(1-\beta)(E_{21} - E_{22}) \\
&= \beta(1-\beta)(\alpha f(n)B - C_{E1} + \alpha J + \gamma N)
\end{aligned} \tag{36}
$$

(3) Public game strategy

The average return of the public to participate in the supervision game strategy is as follows:

$$
\begin{aligned}
E_{31} &= \alpha\beta(M - L) + \beta(1-\alpha)(-L) + \alpha(1-\beta)(M - L - I) + (1-\alpha)(1-\beta)(-L - I) \\
&= \alpha M - L - I + \beta I
\end{aligned} \tag{37}
$$

The average income of the public not participating in the supervision game strategy is as follows:

$$
\begin{aligned}
E_{32} &= \alpha(1-\beta)(-I) + (1-\alpha)(1-\beta)(-I) \\
&= \beta I - I
\end{aligned} \tag{38}
$$

The average expected income of the public is as follows:

$$E_3 = \gamma E_{31} + (1 - \gamma)E_{32} = \alpha\gamma M + \beta\gamma I + \beta I - \gamma L - \beta\gamma I - I \tag{39}$$

The public's copy of the dynamic differential equation is as follows:

$$F(\gamma) = \frac{d\gamma}{dt} = \gamma(E_{31} - E_3) = \gamma(1 - \gamma)(E_{31} - E_{32}) = \gamma(1 - \gamma)(\alpha M - L) \tag{40}$$

2.5.2. The tripartite evolution game stability

Let $F(\alpha) = 0$, $F(\beta) = 0$, and $F(\gamma) = 0$, then, we obtain nine equilibrium points, namely $B_1 = (0,0,0)$, $B_2 = (1,0,0)$, $B_3 = (0,1,0)$, $B_4 = (0,0,1)$, $B_5 = (1,1,0)$, $B_6 = (1,0,1)$, $B_7 = (0,1,1)$, and $B_8 = (1,1,1)$, and then the calculation formula of $B_9$ is as follows:

$$((\alpha, \beta, \gamma) | 0 < \alpha < 1; 0 < \beta < 1; 0 < \gamma < 1)$$

$$\begin{cases} J - C_G - \beta J - \gamma M - \beta f(n)B = 0 \\ \alpha f(n)B - C_{E1} + \alpha J + \gamma N = 0 \\ \alpha M - L = 0 \end{cases}$$

$$\begin{cases} \alpha = \frac{L}{M} \\ \beta = \frac{NJ - NC_G - MC_{E1} + LJ + f(n)BL}{N(J + f(n)B)} \\ \gamma = \frac{MC_{E1} - LJ - f(n)BL}{MN} \end{cases}$$

We can obtain the following formulas from the dynamic equation of the tripartite.

$$\frac{dF(\alpha)}{d\alpha} = (1 - 2\alpha)(J - C_G - \beta J - \gamma M - \beta f(n)B) \tag{41}$$

$$\frac{dF(\beta)}{d\beta} = (1 - 2\beta)(\alpha f(n)B - C_{E1} + \alpha J + \gamma N) \tag{42}$$

$$\frac{dF(\gamma)}{d\gamma} = (1 - 2\gamma)(\alpha M - L) \tag{43}$$

$$\frac{dF(\alpha)}{d\beta} = -\alpha(1 - \alpha)(J + f(n)B) \tag{44}$$

$$\frac{dF(\alpha)}{d\gamma} = -\alpha(1 - \alpha)M \tag{45}$$

$$\frac{dF(\beta)}{d\alpha} = \beta(1 - \beta)(f(n)B + J) \tag{46}$$

$$\frac{dF(\beta)}{d\gamma} = \beta(1 - \beta)N \tag{47}$$

$$\frac{dF(\gamma)}{d\alpha} = \gamma(1 - \gamma)M \tag{48}$$

$$\frac{dF(\gamma)}{d\beta} = 0 \tag{49}$$

According to the theory of stability of evolution game theory, when $\frac{dF(\alpha)}{d\alpha} < 0$, $\frac{dF(\beta)}{d\alpha} < 0$, and $\frac{dF(\gamma)}{d\alpha} < 0$, the stable point of the tripartite subject evolution game is $B_9$.

(1) Gradual equilibrium analysis of the government

According to $J - C_G - \beta J - \gamma M - \beta f(n)B = 0$, that is, $\gamma = \gamma_0 = \frac{J - C_G - \beta J - \beta f(n)B}{M}$. $F(\alpha)$ is close to 0 at this time. Regardless of the probability $\alpha$, the model will be in a stable state. If $\gamma \neq \gamma_0$ then let $F(\alpha) = 0$. Then, we can obtain a stable state $\alpha = 0$, $\alpha = 1$.

When $J - C_G - \beta J - \gamma M - \beta f(n)B < 0$; that is, $\gamma > \gamma_0$, which is because of $M > 0$, $\frac{dF(\alpha)}{d\alpha}\big|_{\alpha=1} > 0$, and $\frac{dF(\alpha)}{d\alpha}\big|_{\alpha=0} < 0$, $\alpha = 0$ is the stable state at this time, and the government tends to choose negative regulatory strategies.

When $J - C_G - \beta J - \gamma M - \beta f(n)B > 0$; that is, $\gamma < \gamma_0$, $\frac{dF(\alpha)}{d\alpha}\big|_{\alpha=1} < 0$, and $\frac{dF(\alpha)}{d\alpha}\big|_{\alpha=0} > 0$, $\alpha = 1$ is the stable state at this time, and the government tends to choose active regulatory strategies.

(2) Gradual equilibrium analysis of the enterprise

If $\alpha f(n)B + \gamma N - C_{E1} + C_{E2} = 0$; that is, $\gamma = \gamma_0 = \frac{C_{E1} - \alpha J - \alpha f(n)B}{N}$, then $F(\beta)$ is close to 0 at this time. Regardless of the probability $\beta$, the model will be in a stable state as a whole.

If $\gamma \neq \gamma_0$ and $F(\beta) = 0$, we can obtain the stable states of $\beta_1 = 0$ and $\beta_2 = 1$.

If $\alpha f(n)B + \gamma N - C_{E1} + \alpha J < 0$; that is, $\gamma < \gamma_0$, we can obtain the result of $\frac{dF(\beta)}{d\beta}\big|_{\beta=1} > 0$, $\frac{dF(\beta)}{d\beta}\big|_{\beta=0} < 0$. $\beta = 0$ is a stable state, and the enterprise tends to implement negative governance pollution.

If $\alpha f(n)B + \gamma N - C_{E1} + \alpha J > 0$; that is, if $\gamma > \gamma_0$, we can obtain the result of $\frac{dF(\beta)}{d\beta}\big|_{\beta=1} < 0$, $\frac{dF(\beta)}{d\beta}\big|_{\beta=0} > 0$. $\beta = 1$ is the stable state, and the enterprise tends to implement active governance pollution.

(3) Gradual equilibrium analysis of the public

If $\alpha M - L = 0$; that is, $\alpha = \alpha_0$, then $F(\gamma)$ unlimited approaches 0, no matter what value $\gamma$ takes, the game is stable.

If $\alpha \neq \alpha_0$ and $F(\gamma) = 0$, we can obtain the stable state of $\gamma_1 = 0$ and $\gamma_2 = 1$.

If $\alpha M - L < 0$; that is, if $\alpha < \alpha_0$, we can obtain the results of $\frac{dF(\gamma)}{d\gamma}\big|_{\gamma=1} > 0$ and $\frac{dF(\gamma)}{d\gamma}\big|_{\gamma=0} < 0$. $\gamma = 0$ is the stable state, and the public tends to not participate in the regulatory strategies.

If $\alpha M - L > 0$; that is, if $\alpha > \alpha_0$, we can obtain the results of $\frac{dF(\gamma)}{d\gamma}\big|_{\gamma=1} < 0$ and $\frac{dF(\gamma)}{d\gamma}\big|_{\gamma=0} > 0$. $\gamma = 1$ is the stable state, and the public tends to participate in the regulatory strategies.

## 3. Results and Discussion

### 3.1. The Results of the Two-Party Game Model

Only when the $DetJ > 0$ and $TrJ < 0$ conditions are met at the same time is the local equilibrium point ESS. Combining the value range of each parameter [45], the four local equilibrium points of $(0,0)$, $(0,1)$, $(1,0)$, and $(1,1)$ will be analyzed below the possibility of ESS.

**Situation 1.** $J < C_G$ and $C_{E1} - J < f(n)B$.

As shown in Table 5, when the punishment received by enterprises for choosing passive pollution control is less than the additional cost to the government of active regulation and when the additional costs incurred by enterprises in actively controlling pollution are less than the sum of the punishment received by enterprises for choosing passive pollution control and the product of a function, then the dependent variable decreases as the independent variable increases and the full maximum subsidy is provided by the government to enterprises; hence, there is one ESS in the Jacobian matrix, namely, $(0,0)$, respectively. The corresponding evolutionary stability strategy is negative supervision and negative pollution control.

**Table 5.** Government and enterprise evolution game balance points in the case of situation one.

| Balance Point | Determinant Value | +/− | Matrix Value | +/− | Stability |
|---|---|---|---|---|---|
| $(0,0)$ | $(J - C_G)(-C_{E1})$ | + | $J - C_G - C_{E1}$ | − | ESS |
| $(0,1)$ | $(-f(n)B - C_G)C_{E1}$ | − | $-f(n)B - C_G + C_{E1}$ | Uncertain | Unstable |
| $(1,0)$ | $(C_G - J)(J - C_{E1} + f(n)B)$ | + | $C_G - C_{E1} + f(n)B$ | + | Unstable |
| $(1,1)$ | $(C_G + f(n)B)(C_{E1} - J - f(n)B)$ | − | $C_G + C_{E1} - J$ | Uncertain | Unstable |
| $(x^*, y^*)$ | $-b_{12}b_{21}$ | + | 0 | 0 | Unstable |

**Situation 2.** $J < C_G$ and $C_{E1} - J > f(n)B$.

As shown in Table 6, when the punishment received by enterprises for choosing passive pollution control is less than the additional cost to the government of active regulation, when the additional costs incurred by enterprises in actively controlling pollution are greater than the sum of the punishment received by enterprises for choosing passive pollution control, and when the product of a function in which the dependent variable decreases as the independent variable increases and the full maximum subsidy is provided by the government to enterprises, there is one ESS in the Jacobian matrix. It is $(0,0)$, respectively, and the corresponding evolutionary stability strategy is negative supervision and negative pollution control.

**Table 6.** Government and enterprise evolution game balance points in the case of situation two.

| Balance Point | Determinant Value | +/− | Matrix Value | +/− | Stability |
|---|---|---|---|---|---|
| $(0,0)$ | $(J - C_G)(-C_{E1})$ | + | $J - C_G - C_{E1}$ | − | ESS |
| $(0,1)$ | $(-f(n)B - C_G)C_{E1}$ | − | $-f(n)B - C_G + C_{E1}$ | Uncertain | Unstable |
| $(1,0)$ | $(C_G - J)(J - C_{E1} + f(n)B)$ | − | $C_G - C_{E1} + f(n)B$ | Uncertain | Unstable |
| $(1,1)$ | $(C_G + f(n)B)(C_{E1} - J - f(n)B)$ | + | $C_G + C_{E1} - J$ | + | Unstable |
| $(x^*, y^*)$ | $-b_{12}b_{21}$ | + | 0 | 0 | Unstable |

**Situation 3.** $J > C_G$ and $C_{E1} - J < f(n)B$.

As shown in Table 7, when the punishment received by enterprises for choosing passive pollution control is greater than the additional cost to the government of active regulation, when the additional costs incurred by enterprises in actively controlling pollution is less than the sum of the punishment received by enterprises for choosing passive pollution control, and when the product of a function in which the dependent variable decreases as the independent variable increases and the full maximum subsidy is provided by the government to enterprises, there is no ESS in the system.

**Table 7.** Government and enterprise evolution game balance points in the case of situation three.

| Balance Point | Determinant Value | +/− | Matrix Value | +/− | Stability |
|---|---|---|---|---|---|
| $(0,0)$ | $(J - C_G)(-C_{E1})$ | − | $J - C_G - C_{E1}$ | Uncertain | Unstable |
| $(0,1)$ | $(-f(n)B - C_G)C_{E1}$ | − | $-f(n)B - C_G + C_{E1}$ | Uncertain | Unstable |
| $(1,0)$ | $(C_G - J)(J - C_{E1} + f(n)B)$ | − | $C_G - C_{E1} + f(n)B$ | Uncertain | Unstable |
| $(1,1)$ | $(C_G + f(n)B)(C_{E1} - J - f(n)B)$ | − | $C_G + C_{E1} - J$ | Uncertain | Unstable |
| $(x^*, y^*)$ | $-b_{12}b_{21}$ | + | 0 | 0 | Unstable |

**Situation 4.** $J > C_G$ and $C_{E1} - J > f(n)B$.

As shown in Table 8, when the punishment received by enterprises for choosing passive pollution control is greater than the additional cost to the government of active

regulation, when the additional costs incurred by enterprises in actively controlling pollution are greater than the sum of the punishment received by enterprises for choosing passive pollution control, and when the product of a function in which the dependent variable decreases as the independent variable increases and the full maximum subsidy is provided by the government to enterprises, there is one ESS in the Jacobian matrix. It is $(1,0)$, respectively, and the corresponding evolutionary stability strategy is active supervision and negative pollution.

**Table 8.** Government and enterprise evolution game balance points in the case of situation four.

| Balance Point | Determinant Value | +/− | Matrix Value | +/− | Stability |
|---|---|---|---|---|---|
| $(0,0)$ | $(J - C_G)(-C_{E1})$ | − | $J - C_G - C_{E1}$ | Uncertain | Unstable |
| $(0,1)$ | $(-f(n)B - C_G)C_{E1}$ | − | $-f(n)B - C_G + C_{E1}$ | Uncertain | Unstable |
| $(1,0)$ | $(C_G - J)(J - C_{E1} + f(n)B)$ | + | $C_G - C_{E1} + f(n)B$ | − | ESS |
| $(1,1)$ | $(C_G + f(n)B)(C_{E1} - J - f(n)B)$ | + | $C_G + C_{E1} - J$ | + | Unstable |
| $(x^*, y^*)$ | $-b_{12}b_{21}$ | + | $0$ | $0$ | Unstable |

According to the above four situations, the following conclusions can be obtained:

(1) There are two ESS points in the game between the government and the enterprise, namely $(0,0)$ and $(1,0)$. The corresponding strategies are negative supervision, negative pollution control, and active supervision, negative pollution.

(2) In situation one and situation two, when the government chooses negative regulatory strategies due to the difference between the cost of active pollution control and the punishment received by the negative pollution control of the enterprise being less than the subsidies for the active pollution control enterprises, enterprises will choose negative pollution control. When enterprises choose negative pollution because the cost of government supervision is higher than the social income from government supervision, the government will choose negative supervision. At this time, the government's and enterprise's evolutionary stability strategies are negative supervision and negative pollution control.

(3) In situation four, when the government chooses to actively regulate the strategy due to the difference between the cost of active pollution control and the punishment received by the negative pollution control of the enterprise being greater than the subsidies for the active pollution control enterprises, enterprises will choose negative pollution control. When enterprises choose negative pollution because the cost of government supervision is less than the social income from government supervision, the government will choose to actively supervise. At this time, the government's and enterprise's evolutionary stability strategy is active supervision and negative pollution control.

*3.2. The Verification of the Two-Party Game Model*

Based on the statistical data of the Chinese government and enterprises in 2022, we assigned values to the initial state of the evolutionary game between the government and enterprises. Let $R_E = 6$, $R_G = 6$, $D = 3$, $C_G = 1.5$, $J = 1$, $C_{E1} = 3$.

Aiming at the stability of the equilibrium point of a two-party evolutionary game, Matlab R2016a has been used to simulate the dynamic evolution process and verify the correctness of the theoretical analysis. The horizontal axis represents time, and the vertical axis represents the probability of the government and business evolutions. According to the results, the simulation evolution diagram of the government and enterprise is obtained, as shown in Figure 2. In the current development situation, industrial pollution regulation is the trend, and choosing to actively regulate enterprises to control industrial pollution has high value for society. Under these conditions, the government is more concerned about how to formulate incentive mechanisms to increase the probability of enterprises actively controlling pollution.

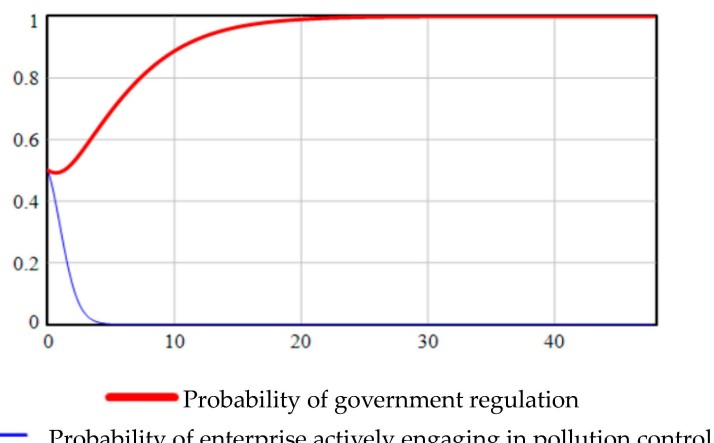

**Figure 2.** Simulation diagram of government and enterprise strategy selection in situation four.

From the simulation conclusion of situation four, the trend speed of enterprise pollution control is influenced by various factors, including the cost of active pollution control by enterprises and government subsidies for choosing active pollution control enterprises. However, if we change the direction of enterprise pollution control, relying solely on the power of the government is not enough. Therefore, we considered introducing another stakeholder, the public, for tripartite evolutionary game analysis.

*3.3. The Results of the Tripartite Game Model*

(1) Government level

According to the gradual, balanced analysis of the government, we can see that $\gamma = \gamma_0 = \frac{J - C_G - \beta J - \beta f(n)B}{M}$. When $C_G$ and $f(n)B$ increase, $\gamma_0$ becomes smaller. That is, the high government supervision cost will cause supervision to become passive. Similarly, when $J$ increases, $\gamma_0$ becomes larger; that is, the greater the social income obtained through supervision, the more active the supervision.

If a local government tends to pursue short-term benefits, environmental management will often be despised. In this case, the government will not take the initiative to encourage enterprises to actively discharge pollution, nor will they take the initiative to provide enterprise subsidies to actively treat pollution. In response to this situation, on the one hand, the government should improve its management system to ensure that while paying attention to economic development, we must not ignore the governance of industrial pollution. We must comprehensively examine the adverse effects of industrial pollution and encourage enterprises to actively treat pollution. On the other hand, the government must also encourage the public to participate in supervision, and at the same time, it is necessary to let enterprises do a good job of prevention and control of industrial pollution.

(2) Enterprise level

It can be seen through $\gamma = \gamma_0 = \frac{C_{E1} - \alpha J - \alpha f(n)B}{N}$, as $N$, $J$, and $f(n)B$ increase, $\gamma_0$ decreases. In other words, the greater the adverse impact of negative pollution control on the enterprise and the more punishment when the company's choice is negative pollution, the more subsidies are given by the government, and the more the enterprise tends to actively treat pollution to avoid corresponding losses. Similarly, when $C_{E1}$ becomes larger, $\gamma_0$ will increase, which indicates that the higher the cost of active pollution control, the more enterprises tend toward negative pollution.

It is found that enterprises will center on their interests and the purpose of profitability and, therefore, lack the initiative to protect the environment, which explains the reason why Chinese industrial pollution control is not high. Therefore, to encourage enterprises to adopt the "active pollution control" strategy, it is necessary to increase the profit margin and adjust the coefficient to make $\alpha f(n)B > C_{E1} - \alpha J - \gamma N$. Therefore, the government can adopt the following measures: Increase the control of enterprises that do not carry out

industrial pollution, issue sufficient allowances to enterprises with industrial pollution, formulate certain punishment measures to improve the efficiency of supervision, encourage the public to assist the government's supervision, and introduce high-tech talents at the same time to improve the country's pollution control technology and minimize the cost of pollution control.

(3) Public level

According to the gradual, balanced analysis of the public, it can be seen that when $L$ increases, $\alpha_0$ will increase, which indicates that when the cost of government supervision is too high, that is, the lower the public income, the more likely the public will participate in supervision. When $M$ increases, A decreases; that is, the more subsidies given by the government when the public participates in supervision, the more the public is more inclined to participate in supervision.

The public's choice of behavior depends on the benefits. Unlike enterprises, pollution will directly affect the quality of life of the public, which is a long-term loss of benefits. In order to enable the public to actively participate in supervision, it is necessary to improve it, pay attention to its benefits, and adjust the relevant parameters [46], $\alpha M - L > 0$. Therefore, the government can take measures to increase the rewards for the public to actively supervise the strategic choice, improve public awareness, and intensify publicity on the consequences caused by enterprises not actively controlling pollution.

Although the government encourages public participation in regulation, public participation in regulation is voluntary. Based on this analysis, it can be concluded that the stable points for the eight equilibrium states are (0,1,1). This means that even when the government does not regulate, enterprises will choose the "active pollution control" strategy with public participation in regulation, which is a long-term goal and inevitable trend for the development of China's industrial industry.

*3.4. The Simulation Analysis of the Tripartite Game Model*

Through the analysis of the evolution theory of the government, the public, and enterprise, it can be concluded that (0,1,1) is the equilibrium point. We assigned values to the subjects of the tripartite evolutionary game, letting $R_E = 6$, $R_G = 6$, $D = 3$, $C_G = 4$, $J = 3$, $C_{E1} = 3$, $B = 5$, $L = 1.5$, $M = 1$, $N = 6$, and $I = 4$.

To clearly illustrate the evolution paths of the government, the enterprise, and the public, Matlab R2016a has been used to simulate the dynamic evolution process. The horizontal axis represents time, and the vertical axis represents the probability of government, enterprises, and the public choosing. According to the results, the simulation evolution diagram of government, enterprise, and the public is obtained, as shown in Figure 3. From the above analysis, it can be seen that by introducing the public to pollution control, the government can achieve positive pollution control effects for enterprises.

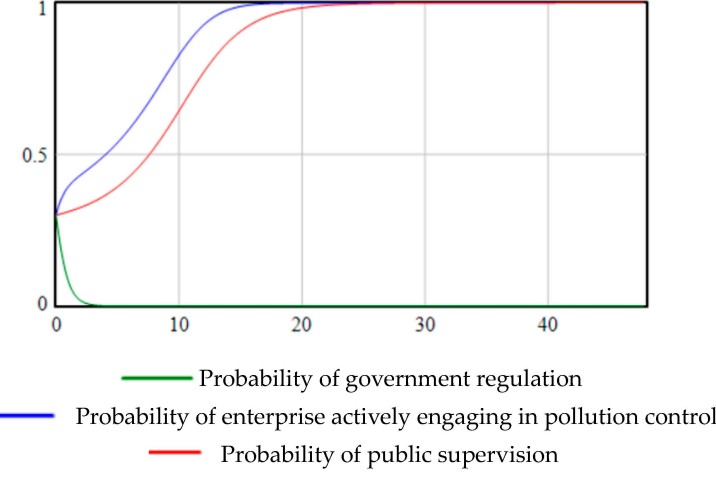

Probability of government regulation

Probability of enterprise actively engaging in pollution control

Probability of public supervision

**Figure 3.** Simulation analysis of the tripartite evolutionary game.

## 4. Conclusions

In this study, taking public participation as an important factor in industrial pollution control into a unified analytical framework, we put forward a two-party game model of government–enterprise and a tripartite game model of government–enterprise–public and explored the multi-agent behavior strategy path of industrial pollution control. According to the economic development situation of China, the government–enterprise and government–enterprise–public game models were assigned values and simulated in scenarios. The main conclusions are as follows:

(1) The government needs to adjust the pollution supervision model. It is not feasible to rely solely on government pollution regulation. It is necessary to add public participation based on a single government-led regulatory model, gradually reform the government regulatory model, and make pollution regulation more efficient. Joining public participation in collaborative regulation is an important breakthrough in solving industrial pollution.

(2) The enterprises strive to reduce the cost of pollution control. Enterprises evade pollution control mainly to reduce production costs and prevent the weakening of their market competitiveness due to increased costs. If enterprises obtain national preferential policies to obtain environmental protection technology and equipment, the cost of pollution control will be significantly reduced, and the tangible and intangible benefits, such as the economic effects and reputation obtained by enterprises, will be higher than the cost of pollution control. The awareness of enterprises and government encouragement policies are of great significance for the development of industrial industries.

(3) There is a positive relationship between the power of public supervision and the active pollution control of enterprises. When enterprises do not actively control pollution and thus create negative impacts on the social environment, the supervision power of the public plays a decisive role. The public adopts regulatory strategies to restrain enterprises in pollution output and urges enterprises to achieve positive pollution control effects.

This paper only discusses three stakeholders, i.e., the government, the public, and enterprises. There are many more stakeholders involved in industrial pollution. We will establish a multi-participant strategy model to study the influence of other participants on industrial pollution control in the future.

**Author Contributions:** Conceptualization, N.Y.; methodology, N.Y. and M.L.; software, M.L.; validation, N.Y. and M.L.; formal analysis, N.Y.; writing—original draft preparation, N.Y. and M.L.; writing—review and editing, N.Y.; supervision, N.Y.; funding acquisition, N.Y. All authors have read and agreed to the published version of the manuscript.

**Funding:** This research was funded by the Anhui Province Social Science Innovation and Development Research Project, grant number 2023CX039, the Support Program for Outstanding Young Talents in Universities, grant number gxyq2022070, and the Excellent Research and Innovation Teams in Anhui Province's Universities, grant number 2022AH010097.

**Institutional Review Board Statement:** Not applicable.

**Informed Consent Statement:** Not applicable.

**Data Availability Statement:** The data are available upon request from the corresponding author.

**Conflicts of Interest:** The authors declare no conflict of interest.

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
