# Peer review of "Analysis of the Dynamic Evolution Game of Government, Enterprise and the Public to Control Industrial Pollution"

_sustainability, doi:10.3390/su16072760_

Round 1

Reviewer 1 Report

Comments and Suggestions for Authors

The manuscript presents an approach to understanding the interplay between government, businesses, and the public in the context of industrial pollution control. The use of evolutionary game theory to model these interactions is potentially valuable. However, there are several areas where the paper could be significantly improved.

1. The subject matter is highly relevant and timely, particularly the inclusion of public participation in the model of industrial pollution governance. The theoretical framework appears original, but a clearer connection with existing literature would strengthen the paper.

2. The evolutionary game model involving government, enterprises, and the public is theoretically sound. However, the manuscript needs to provide a clearer explanation of the model construction, including assumptions and parameter selection.

3. The stability analysis of the model requires more detailed exposition for better comprehension and assessment by the readers.

4. The paper provides analysis of potential stable states under various scenarios but lacks empirical data for validation. Incorporating case studies or real-world data to corroborate the model predictions would be beneficial.

5. The discussion section needs further development, especially in elucidating the practical implications of the research findings for real-world industrial pollution control strategies. Also, limitations of the current study and directions for future research should be discussed.

6. Consider incorporating more real-world policies and historical cases to support the arguments about the roles of government, businesses, and the public in pollution control.

7. Given the rapid developments in this field, the authors are advised to reference and cite more recent studies to ensure the timeliness and relevance of the paper.

8. Some figures should be added to clarify the results

Comments on the Quality of English Language

Extensive editing of English language required

Author Response

1.The subject matter is highly relevant and timely, particularly the inclusion of public participation in the model of industrial pollution governance. The theoretical framework appears original, but a clearer connection with existing literature would strengthen the paper.

Response: The authors sincerely thank the reviewer for the positive comments and the suggestions on improving the manuscript. Following the reviewer's suggestions, the authors have carefully summarized the existing literature. The revised part has been marked in red color in the revision.

2.The evolutionary game model involving government, enterprises, and the public is theoretically sound. However, the manuscript needs to provide a clearer explanation of the model construction, including assumptions and parameter selection.

Response: The authors sincerely thank the reviewer for the positive comments and the suggestions on improving the manuscript. Following the reviewer's suggestions, the authors have carefully provided a clearer explanation of assumptions and parameter selection. All the modifications are printed in red in the revised manuscript .

The modifications to the two parties evolutionary game model are as follows:

Hypothesis 1(H1): Participating subject. In the evolutionary game model, there are two participants, one is government departments, and the other is an enterprise.

Hypothesis 2(H2): Cooperative strategy. Assuming that the government has the right to control the enterprises, so as to improve the indicators of the enterprises' emissions to the level of environmental protection, and the pollution emitted by the enterprises will not affect the lives of the public and the environment. It is also possible that the government only pays attention to one-sided economic development and ignores the protection of the ecological environment and public interests. Enterprises can choose to actively control pollution, which not only meets environmental protection standards but also minimizes pollution. Enterprises can also choose not to actively control pollution , that is, passive pollution control, they can only meet the minimum standards of pollution control, ignoring the actual needs of the public. Therefore, both sides of the game have two strategies, the government's strategy is (active regulation, negative regulation), and the enterprise's strategy is (active pollution control, negative pollution control). The strategic space of this game is: (active regulation, positive pollution control), (active regulation, negative pollution control), (negative regulation, active pollution control), (negative regulation, negative pollution control).

Hypothesis 3(H3): Government benefits and cost. Both parties of the game are rational economics with their own interests, and all the benefits, costs and expenses in the model can be quantified. The government's choice of active regulation and negative regulation strategy mainly depends on the government's profit expectations. The process of enterprise pollution control involves basic enterprise revenue, regulatory costs and incentives. The difference between the government’s income and regulatory costs during passive regulation can be expressed by , and the additional cost of government active regulation can be expressed by . If the enterprise is passive in pollution control, the government will bear the corresponding treatment costs, assuming that it is expressed by . Government incentive measures mean that if enterprises actively control pollution, the government will give certain subsidies. The subsidy will be issued according to the "regression" mechanism, that is, the subsidy amount in the process of industrial pollution supervision will decrease with time. Assuming that the enterprise chooses to actively control pollution, then the subsidy issued by the government to the enterprise is , where is a decreasing function, . The government's full maximum subsidy can be expressed as , the subsidy uses the "regression" mechanism to enable enterprises to obtain subsidies in batches. During this period, enterprises must actively control pollution to obtain subsidies at each stage.

Hypothesis 4(H4): Enterprise benefits and cost. The enterprise has two strategic options (active pollution control, negative pollution control). If the enterprise chooses to actively control pollution, the difference between the normal income of the enterprise and the cost of pollution control is , the additional cost of the enterprise is , and the subsidy received by the enterprise is . If the enterprise chooses passive pollution control, the penalty received is recorded as .

The modifications to the three parties evolutionary game model are as follows:

Hypothesis 1(H1): Government strategy. The government is facing two strategies (active supervision and negative supervision). If the government takes active supervision, it is necessary to pay additional cost . The difference between the normal income and regulatory cost of the government during negative supervision is . When the enterprise chooses negative pollution control, the government's governance cost is . The government gives the subsidies for active pollution control enterprises to be , and the government's subsidy to the public is .

Hypothesis 2(H2): Enterprise strategy. The enterprise faces two strategies (active pollution control, negative pollution control). Assume that the difference between the normal income and the cost of pollution control when the enterprise implements negative pollution is . When the enterprise spends additional costs when actively treating pollution, it is recorded as .When choosing negative pollution control, the punishment by the enterprise is represented by . The loss of non -active pollution control to enterprises is recorded as .

Hypothesis 3(H3): Public strategy. The public face two strategies (participating in supervision , not participating in supervision). Under the supervision of the public's participation, the cost of its supervision is , and When the public participates in supervision, the government's subsidy is . If it is not active in pollution control, the harm of the public will be recorded as .

Hypothesis 4(H4): Strategy probability. The proportion of the government's choice of active supervision is , and the proportion of enterprise selection of active pollution is , and the proportion of the public chose to participate in the supervision is . Among them, ,,.

3.The stability analysis of the model requires more detailed exposition for better comprehension and assessment by the readers.

Response: Thanks for the comments on the value of this manuscript. The stability of the model has been elaborated in more detail. The detailed information is marked in red color in the revision.

4.The paper provides analysis of potential stable states under various scenarios but lacks empirical data for validation. Incorporating case studies or real-world data to corroborate the model predictions would be beneficial.

Response: The authors sincerely thank the reviewer for the positive comments and the suggestions on improving the manuscript. Following the reviewer's suggestions, the authors have carefully added section 2.5 and section 3.3 as empirical data for validation. All the modifications are printed in red in the revised manuscript.

5.The discussion section needs further development, especially in elucidating the practical implications of the research findings for real-world industrial pollution control strategies. Also, limitations of the current study and directions for future research should be discussed.

Response: The authors appreciated the professional suggestion from the reviewer. The further development has been discussed, especially limitations of the current study and directions for future research has been added. All the modifications are printed in red in the revised manuscript and provided as follows:

With the development of industry, environmental pollution has also emerged. Today, humans have made significant improvements in material terms, but they have paid the price of enduring environmental pollution. With the rapid development of the economy and society, there has been a serious conflict between the increasingly serious ecological and environmental problems and the growing environmental demand of people, which has led to the country increasing its control of industrial pollution. So, we should gradually pay attention to the problems caused by industrial pollution. However, due to the long-term reliance on the development model that comes at the cost of the environment, it has resulted in huge costs of path dependence and transformation, making it difficult for the country to effectively implement control over industrial pollution. Nowadays, many enterprises can only achieve emission standards, but the pollutants they discharge still have a negative impact on public life and the environment. However, relying solely on government efforts for prevention and control is far from enough. Therefore, in the case of market failure caused by environmental externalities, in order to ensure the well-being of society, it is necessary to introduce a new and effective regulatory method - public participation in regulation.

However, in China's government regulatory policies, there are still problems such as incomplete regulatory mechanisms and unclear allocation of rights and responsibilities, which have led to low regulatory efficiency in relevant departments and poor pollution control status of enterprises.

The main research conclusions of this study are as follows:

(1) The government needs to adjust the pollution supervision model. It is not feasible to rely solely on government pollution regulation. It is necessary to add public participation on the basis of a single government led regulatory model, gradually reform the government regulatory model, and make pollution regulation more efficient. Joining public participation in collaborative regulation is an important breakthrough in solving industrial pollution. On the one hand, the government is increasing subsidies for enterprises engaged in industrial pollution supervision. On the other hand, the government is formulating and promoting an industrial pollution control standard system and improving national industrial technology to achieve a virtuous cycle in the industrial industry. At the same time, the government establishes and improves a public access information system to save public regulatory costs, and vigorously promotes the awareness of environmental protection.

(2) Strive to reduce the cost of enterprise pollution control. Enterprises evade pollution control mainly to reduce production costs and prevent the weakening of their market competitiveness due to increased costs. If enterprises obtain national preferential policies to obtain environmental protection technology and equipment, the cost of pollution control will be significantly reduced, and the tangible and intangible benefits such as economic effects and reputation obtained by enterprises will be higher than the cost of pollution control. The awareness of enterprises and government encouragement policies are of great significance for the development of industrial industries. If the benefits of active pollution control by enterprises are less than the benefits of passive pollution control by enterprises, the government must have strong incentive policies to encourage enterprises to actively control pollution. With the participation of the public in regulation, if active pollution control is not carried out, enterprises will suffer certain losses. In order to maintain their own reputation and economic interests, enterprises will also choose to actively control pollution.

(3) Actively enhance the public's willingness to actively participate. The public's regulatory capacity is uneven, and there is a lack of professional knowledge and legal regulations. For the public, self-interest is the most important. However, in this society, the public is directly affected by the environment. When a enterprise does not actively control pollution and brings negative impacts to the social environment, that is to say, when there is a conflict of rights and interests with the public, people will definitely adopt a regulatory strategy to constrain the discharge of pollutants by the company. In this way, society needs to strengthen professional knowledge training for the public, systematically learn various laws, regulations and professional knowledge, and improve the efficiency of public pollution supervision work.

Although some conclusions were drawn in the end of the paper, due to our research level and time, there are still some limitations in the paper. We will improve it in the following aspects:

(1) Add polluter to the model. Although only the three stakeholders of government, the public, and enterprise were discussed in the paper, there are far more stakeholders involved in industrial pollution in real life. Pollutant polluters can be included. Based on this, a strategy model for four participants can be established to analyze the impact of pollutant polluters on the selection strategies of other participants.

(2) Add government propaganda factors to the model. Because many of the public still value short-term benefits, it is crucial to make them aware of the long-term environmental gains and losses. The government should dispatch a professional propaganda team to gradually infiltrate the public's thinking, making environmental protection an active part of public participation. This will weaken the passivity of public participation and make industrial pollution supervision work easier.

6.Consider incorporating more real-world policies and historical cases to support the arguments about the roles of government, businesses, and the public in pollution control.

Response: Thanks for the comments on the value of this manuscript. The current situation of public participation in China has increased. The detailed information is marked in red color in the revision.

7.Given the rapid developments in this field, the authors are advised to reference and cite more recent studies to ensure the timeliness and relevance of the paper.

Response: The authors sincerely thank the reviewer for the positive comments and the suggestions on improving the manuscript. Following the reviewer's suggestions, the authors have carefully revised the manuscript.

8.Some figures should be added to clarify the results.

Response: The authors appreciated the professional suggestion from the reviewer. Following the reviewer's suggestions, the authors have carefully added some figures to clarify the results. All the modifications are printed in red in the revised manuscript.

Reviewer 2 Report

Comments and Suggestions for Authors

Manuscript ID: sustainability-2803596. Title: Analysis of the dynamic evolution game of government, enterprise and the public to control industrial pollution.

Comments:

1. Please include the authors' email addresses.

2. The abstract should be significantly improved. Please include the technical methodology used and the main results obtained. It would be useful to support the results shown with quantitative information.

3. Please show in the abstract clearly and concisely the objective of this study.

4. What is the place of study? Evaluate the convenience of including it in the title. In the abstract if it is necessary to include it.

5. Please adjust the references to the journal format.

6. Please write the entire manuscript in the third person. For example, see L28.

7. Sections 1 and 2 should be integrated. This information corresponds to the introduction.

8. Please remove extra spaces between words. For example, see L55.

9. Please show in the introduction of the manuscript the practical utility of this study.

10. Section 3 can also be integrated with sections 1 and 2. This information corresponds to the introduction.

11. Section 3.1. corresponds to a conventional chapter on materials and methods. Please evaluate.

12. Please include a figure with a flow chart to visualize the methodology used in this study.

13. Section 3.3. is very difficult to follow. Authors should make every effort to provide greater clarity and simplicity.

14. Likewise, section 4 is very dense. The authors should make an effort to present all the information included in a practical way. 

15. This article does not show the simulation of the possible scenarios to be considered. Which software was used?

16. What was the method used to validate the results of this study?

17. Overall, this article must be significantly improved in order to be accepted by a high level journal. I suggest using a conventional structure: introduction, materials and methods, results, discussion, and conclusions. Please contrast your results with other authors.

18. The first paragraph of the conclusions is very general.

19. All conclusions are very general. Please give more relevance to the conclusions.

20. More technical depth is needed in this article. Initially, I suggest rejecting the article. However, it is advisable to give the authors a second chance to significantly improve the paper. Then, I will be able to give a definitive concept.

Comments on the Quality of English Language

6. Please write the entire manuscript in the third person. For example, see L28.

8. Please remove extra spaces between words. For example, see L55.

Author Response

1.Please include the authors' email addresses.

Response: The authors sincerely thank the reviewer for the positive comments and the suggestions on improving the manuscript. The authors' email addresses has been added.

2.The abstract should be significantly improved. Please include the technical methodology used and the main results obtained. It would be useful to support the results shown with quantitative information.

Response: The authors appreciated the professional suggestion from the reviewer. . Following the reviewer's suggestions, the authors have carefully improved the abstract. All the modifications are printed in red in the revised manuscript and provided as follows:

With the continuous development of industrialization, high -intensity energy consumption has made regional environmental pollution problems very prominent. Promoting coordinated and efficient pollution control can promote comprehensive green transformation of economic and social development, and is a key path to forming new quality productive forces. First of all, in response to China's current industrial development and problems, the government and enterprises are selected as the participating entity to build a game model of industrial pollution governance. Secondly, in view of the necessity of public participation in the current situation, we build a model of industrial pollution governance evolutionary game models involved in the public, government, and enterprises, and analyze the different strategies and evolutionary stability points of various acts. According to the current situation of China's economic development, the parameters of the two parties and three parties evolutionary game are assigned values and simulated for analysis. Finally, the public participation through the analysis has become an indispensable force in the current industrial pollution supervision, and feasible suggestions such as creating a favorable environment and promoting the development of the work industry are proposed.

3.Please show in the abstract clearly and concisely the objective of this study.

Response: Thanks for the comments on the value of this manuscript. The objective of this study has been shown in the abstract. The detailed information is marked in red color in the revision.

4.What is the place of study? Evaluate the convenience of including it in the title. In the abstract if it is necessary to include it.

Response: The authors sincerely thank the reviewer for the positive comments and the suggestions on improving the manuscript. In the evolutionary game model of industrial pollution regulation, a comparison was made between models with and without public participation to identify the significance and necessity of public participation. This section is elaborated in the abstract, which provides a detailed explanation of the title.

5.Please adjust the references to the journal format.

Response: The authors sincerely thank the reviewer for the positive comments and the suggestions on improving the manuscript. Following the reviewer's suggestions, the authors have carefully revised the references. All the modifications are printed in red in the revised manuscript.

6.Please write the entire manuscript in the third person. For example, see L28.

Response: Thanks for the insight comments on the value of this manuscript. We are very sorry for our fault, we have revised the entire manuscript. All the revised part has been marked in red color in the revision.

7.Sections 1 and 2 should be integrated. This information corresponds to the introduction.

Response: The authors sincerely thank the reviewer for the positive comments and the suggestions on improving the manuscript. Sections 1 and 2 have be integrated.

8.Please remove extra spaces between words. For example, see L55.

Response: Thanks for the comments on the value of this manuscript. Extra spaces between words have been removed.

9.Please show in the introduction of the manuscript the practical utility of this study.

Response: The authors appreciated the professional suggestion from the reviewer. The manuscript the practical utility of this study has been added. The revised part has been marked in red color in the revision.

10.Section 3 can also be integrated with sections 1 and 2. This information corresponds to the introduction.

Response: The authors sincerely thank the reviewer for the positive comments. Sections 1 and 2 have be integrated.

11.Section 3.1. corresponds to a conventional chapter on materials and methods. Please evaluate.

Response: The authors appreciated the professional suggestion from the reviewer. . Following the reviewer's suggestions, the origin of evolutionary game method and the difference between traditional games has increased. All the modifications are printed in red in the revised manuscript.

12.Please include a figure with a flow chart to visualize the methodology used in this study.

Response: The authors appreciated the professional suggestion from the reviewer. Following the reviewer's suggestions, the flow chart has been included and shown in Figure 1.

13.Section 3.3. is very difficult to follow. Authors should make every effort to provide greater clarity and simplicity.

Response: The authors appreciated the professional suggestion from the reviewer. Following the reviewer's suggestions, the authors have carefully revised section 3.3. All the modifications are printed in red in the revised manuscript.

14.Likewise, section 4 is very dense. The authors should make an effort to present all the information included in a practical way.

Response: The authors sincerely thank the reviewer for the positive comments. Sections 4 has been modified, all the revised part has been marked in red color in the revision.

15.This article does not show the simulation of the possible scenarios to be considered. Which software was used?

Response: The authors appreciated the professional suggestion from the reviewer. Matlab R2016a has been used to simulate.

16.What was the method used to validate the results of this study?

Response: The authors sincerely thank the reviewer for the positive comments and the suggestions on improving the manuscript. Based on the statistical data of China government and enterprises in 2022, in order to validate the results of this study, the authors have carefully added simulation analysis of evolutionary game between two and three parties. All the modifications are printed in red in the revised manuscript.

17.Overall, this article must be significantly improved in order to be accepted by a high level journal. I suggest using a conventional structure: introduction, materials and methods, results, discussion, and conclusions. Please contrast your results with other authors.

Response: The authors sincerely thank the reviewer for the positive comments and the suggestions on improving the manuscript. Section 1 is introduction, section 2 is evolutionary game analysis of two parties, section 3 is evolutionary game analysis of tripartite, section 4 is conclusion.

18.The first paragraph of the conclusions is very general.

Response: The authors appreciated the professional suggestion from the reviewer. . Following the reviewer's suggestions, the authors have carefully supplemented and elaborated on the first paragraph of the conclusion. All the modifications are printed in red in the revised manuscript.

19.All conclusions are very general. Please give more relevance to the conclusions.

Response: Thanks for the comments on the value of this manuscript. All conclusions have been supplemented in detail. All the modifications are printed in red in the revised manuscript and provided as follows:

With the development of industry, environmental pollution has also emerged. Today, humans have made significant improvements in material terms, but they have paid the price of enduring environmental pollution. With the rapid development of the economy and society, there has been a serious conflict between the increasingly serious ecological and environmental problems and the growing environmental demand of people, which has led to the country increasing its control of industrial pollution. So, we should gradually pay attention to the problems caused by industrial pollution. However, due to the long-term reliance on the development model that comes at the cost of the environment, it has resulted in huge costs of path dependence and transformation, making it difficult for the country to effectively implement control over industrial pollution. Nowadays, many enterprises can only achieve emission standards, but the pollutants they discharge still have a negative impact on public life and the environment. However, relying solely on government efforts for prevention and control is far from enough. Therefore, in the case of market failure caused by environmental externalities, in order to ensure the well-being of society, it is necessary to introduce a new and effective regulatory method - public participation in regulation.

However, in China's government regulatory policies, there are still problems such as incomplete regulatory mechanisms and unclear allocation of rights and responsibilities, which have led to low regulatory efficiency in relevant departments and poor pollution control status of enterprises.

The main research conclusions of this study are as follows:

(1) The government needs to adjust the pollution supervision model. It is not feasible to rely solely on government pollution regulation. It is necessary to add public participation on the basis of a single government led regulatory model, gradually reform the government regulatory model, and make pollution regulation more efficient. Joining public participation in collaborative regulation is an important breakthrough in solving industrial pollution. On the one hand, the government is increasing subsidies for enterprises engaged in industrial pollution supervision. On the other hand, the government is formulating and promoting an industrial pollution control standard system and improving national industrial technology to achieve a virtuous cycle in the industrial industry. At the same time, the government establishes and improves a public access information system to save public regulatory costs, and vigorously promotes the awareness of environmental protection.

(2) Strive to reduce the cost of enterprise pollution control. Enterprises evade pollution control mainly to reduce production costs and prevent the weakening of their market competitiveness due to increased costs. If enterprises obtain national preferential policies to obtain environmental protection technology and equipment, the cost of pollution control will be significantly reduced, and the tangible and intangible benefits such as economic effects and reputation obtained by enterprises will be higher than the cost of pollution control. The awareness of enterprises and government encouragement policies are of great significance for the development of industrial industries. If the benefits of active pollution control by enterprises are less than the benefits of passive pollution control by enterprises, the government must have strong incentive policies to encourage enterprises to actively control pollution. With the participation of the public in regulation, if active pollution control is not carried out, enterprises will suffer certain losses. In order to maintain their own reputation and economic interests, enterprises will also choose to actively control pollution.

(3) Actively enhance the public's willingness to actively participate. The public's regulatory capacity is uneven, and there is a lack of professional knowledge and legal regulations. For the public, self-interest is the most important. However, in this society, the public is directly affected by the environment. When a enterprise does not actively control pollution and brings negative impacts to the social environment, that is to say, when there is a conflict of rights and interests with the public, people will definitely adopt a regulatory strategy to constrain the discharge of pollutants by the company. In this way, society needs to strengthen professional knowledge training for the public, systematically learn various laws, regulations and professional knowledge, and improve the efficiency of public pollution supervision work.

Although some conclusions were drawn in the end of the paper, due to our research level and time, there are still some limitations in the paper. We will improve it in the following aspects:

(1) Add polluter to the model. Although only the three stakeholders of government, the public, and enterprise were discussed in the paper, there are far more stakeholders involved in industrial pollution in real life. Pollutant polluters can be included. Based on this, a strategy model for four participants can be established to analyze the impact of pollutant polluters on the selection strategies of other participants.

(2) Add government propaganda factors to the model. Because many of the public still value short-term benefits, it is crucial to make them aware of the long-term environmental gains and losses. The government should dispatch a professional propaganda team to gradually infiltrate the public's thinking, making environmental protection an active part of public participation. This will weaken the passivity of public participation and make industrial pollution supervision work easier.

20.More technical depth is needed in this article. Initially, I suggest rejecting the article. However, it is advisable to give the authors a second chance to significantly improve the paper. Then, I will be able to give a definitive concept.

Response: The authors sincerely thank the reviewer for the positive comments and the suggestions on improving the manuscript. All the modifications are printed in red in the revised manuscript.

Reviewer 3 Report

Comments and Suggestions for Authors

The manuscript entitled "Analysis of the Dynamic Evolution Game of Government, Enterprise, and the Public to Control Industrial Pollution" by Na Yu and Meilin Lu attempts to construct a game-theoretic model involving the government, enterprises, and the public in the context of industrial pollution management in China. The paper is motivated by the escalating environmental challenges posed by industrialization and the need for more efficient governance strategies in pollution control. However, the paper would benefit from:

1. It lacks a clear statement of the research question or hypotheses in Introduction. Clarification of these aspects would enhance the reader's understanding of the study's objectives(Lin22-42).

2. The review could be improved by directly linking these studies to the specific context of the game-theoretic approach proposed in this paper. An assessment of the gaps in the existing literature that this research aims to fill would be beneficial(Lin43-145).

3. The analysis of the game results is detailed, presenting various equilibrium points and their implications for policy and strategy. However, the discussion in Result and Discussion could be enhanced by providing more context-specific interpretations of these results. For example, how do these findings relate to current policies in China? How might they inform future governance strategies for industrial pollution?

4.Lin 7-8 “enterprises need to further strengthen sewage discharge to achieve sustainable industrial development” Does this statement imply that enterprises need to emit more waste? If so, which subsequent article can support this viewpoint?

5. Lin 28 Suggest replacing 'my country' with 'China'

6. The results of the paper are obvious, including the relatively obvious roles of the government, enterprises, and public participation. Taking public participation as an example, its effectiveness is related to various factors such as education. What is the current state of public participation in China, and is it possible to examine the impact of the level of public participation in this paper?

Comments on the Quality of English Language

The language of the paper requires further refinement and modification.

Author Response

1.It lacks a clear statement of the research question or hypotheses in Introduction. Clarification of these aspects would enhance the reader's understanding of the study's objectives(Lin22-42).

Response: The authors appreciated the professional suggestion from the reviewer. The research question has been stated. The revised part has been marked in red color in the revision.

2.The review could be improved by directly linking these studies to the specific context of the game-theoretic approach proposed in this paper. An assessment of the gaps in the existing literature that this research aims to fill would be beneficial(Lin43-145).

Response: The authors sincerely thank the reviewer for the positive comments and the suggestions on improving the manuscript. Following the reviewer's suggestions, the authors have carefully revised the manuscript. The revised part has been marked in red color in the revision.

3.The analysis of the game results is detailed, presenting various equilibrium points and their implications for policy and strategy. However, the discussion in Result and Discussion could be enhanced by providing more context-specific interpretations of these results. For example, how do these findings relate to current policies in China? How might they inform future governance strategies for industrial pollution?

Response: The authors sincerely thank the reviewer for the positive comments and the suggestions on improving the manuscript. Based on the statistical data of China government and enterprises in 2022, in order to validate the results of this study, the authors have carefully added simulation analysis of evolutionary game between two and three parties. All the modifications are printed in red in the revised manuscript.

4.Lin 7-8 “enterprises need to further strengthen sewage discharge to achieve sustainable industrial development” Does this statement imply that enterprises need to emit more waste? If so, which subsequent article can support this viewpoint?

Response: Thanks for the insight comments on the value of this manuscript. We are very sorry for our fault, we have revised the lin 7-8. The detailed information is marked in red color in the revision.

5.Lin 28 Suggest replacing 'my country' with 'China'

Response: Thanks for the insight comments on the value of this manuscript. the authors have carefully revised the manuscript. The detailed information is marked in red color in the revision.

6.The results of the paper are obvious, including the relatively obvious roles of the government, enterprises, and public participation. Taking public participation as an example, its effectiveness is related to various factors such as education. What is the current state of public participation in China, and is it possible to examine the impact of the level of public participation in this paper?

Response: Thanks for the comments on the value of this manuscript. The current situation of public participation in China has increased. The detailed information is marked in red color in the revision and provided as follows:

In the context of dual carbon, the public has put forward new and higher-level requirements for sustainable development and environmental protection. Therefore, regulating industrial pollution has become an urgent task. With the downward shift of the focus of social governance, in recent years, various regions have organically combined grassroots conflict mediation, resulting in many practical innovations that can improve the level of public management. At the same time, it has also provided some reference areas for the development of the industrial pollution supervision industry. By various means, we can improve the level of public participation in supervision, fully mobilize the enthusiasm of grassroots supervision, build a regulatory community.

However, there are some problems for the public in pollution regulation. For example, with the continuous improvement of China's industrial pollution supervision system, the current laws, regulations, and rules can no longer meet the practical needs of industrial pollution supervision, nor can they meet the practical needs of public participation in supervision. In the reality of public participation in industrial pollution regulation, due to their weak subjective will and unclear boundaries of rights and responsibilities, their initiative and enthusiasm to participate in industrial pollution regulation are greatly reduced.

Round 2

Reviewer 1 Report

Comments and Suggestions for Authors

This manuscript can be accepted.

Author Response

The authors sincerely thank the reviewer for the positive comments and the suggestions on improving the manuscript.

Reviewer 2 Report

Comments and Suggestions for Authors

Manuscript ID: sustainability-2803596-R2. Title: Analysis of the dynamic evolution game of government, enterprise and the public to control industrial pollution.

Comments:

1. The abstract should be significantly improved. Please include the technical methodology used and the main results obtained. It would be useful to support the results shown with quantitative information.

2. Please show in the abstract clearly and concisely the objective of this study.

3. Section 3. corresponds to a conventional chapter on materials and methods.

4. This article does not show the simulation of the possible scenarios to be considered. Which software was used? Include in detail in the article.

5. What was the method used to validate the results of this study? Include in detail in the article.

6. Overall, this article must be significantly improved in order to be accepted by a high level journal. I suggest using a conventional structure: introduction, materials and methods, results, discussion, and conclusions.

7. In this new version, the conclusions are very extensive. Please synthesize.

Author Response

1. The abstract should be significantly improved. Please include the technical methodology used and the main results obtained. It would be useful to support the results shown with quantitative information.

Response: The authors sincerely thank the reviewer for the positive comments and the suggestions on improving the manuscript. The abstract has been revised. All the modifications are printed in red in the revised manuscript.

2. Please show in the abstract clearly and concisely the objective of this study.

Response: The authors appreciated the professional suggestion from the reviewer. The abstract has been revised. All the modifications are printed in red in the revised manuscript and provided as follows:

This paper constructs a two-party evolutionary game model of government and enterprise to solve the dilemma of industrial pollution control and explore the mode of government and enterprise collaborative governance. The local equilibrium points of the game model in four cases are calculated and analyzed, the results show that government power lonely can not promote enterprises to achieve the ideal pollution discharge effect, and it is necessary to introduce public power for supervision. Based on this, a tripartite evolutionary game model of government, public and enterprise is proposed. When the costs and benefits of the tripartite game players meet certain conditions, the system will evolve in an equilibrium state (0,1,1). According to the current situation of China's economic development, the parameters of the two-party and tripartite evolutionary game are assigned, and the operating path and system evolution trajectory of the two-party and tripartite industrial pollution control are simulated by Matlab R2016a software. It is indicated that whether the government participates in supervision or not, the enterprise will actively control pollution under the public strong supervision, which can provide feasible suggestions for the selection of industrial pollution control policies.

3. Section 3. corresponds to a conventional chapter on materials and methods.

Response: The authors sincerely thank the reviewer for the positive comments and the suggestions on improving the manuscript. Section 2 is materials and method, and section 3 is results and discussion. The detailed information is marked in red color in the revision.

4. This article does not show the simulation of the possible scenarios to be considered. Which software was used? Include in detail in the article.

Response: Thanks for the comments on the value of this manuscript. Matlab R2016a has been used to simulate. The detailed information is marked in red color in the revision.

5.What was the method used to validate the results of this study? Include in detail in the article.

Response: The authors sincerely thank the reviewer for the positive comments and the suggestions on improving the manuscript. Following the reviewer's suggestions, the authors have carefully added the method. All the modifications are printed in red in the revised manuscript.

6. Overall, this article must be significantly improved in order to be accepted by a high level journal. I suggest using a conventional structure: introduction, materials and methods, results, discussion, and conclusions.

Response: The authors appreciated the professional suggestion from the reviewer. The structure has been revised. All the modifications are printed in red in the revised manuscript and provided as follows:

  1. Introduction
  2. Materials and method

2.1. The two-party game model assumption

2.2. The two-party game model

2.3. The two-party game model analysis

2.3.1. The government game model solution

2.3.2. The enterprise game model solution

2.3.3. The two-party game model solution

2.4. The tripartite game model assumption

2.5. The tripartite game model

2.5.1. The dynamic copy equation

2.5.2. The tripartite evolution game stability

  1. Results and discussion

3.1. The results of the two-party game model

3.2. The verification of the two-party game model

3.3. The results of the tripartite game model

3.4. The simulation analysis of the tripartite game model

  1. Conclusions

7.In this new version, the conclusions are very extensive. Please synthesize.

Response: The authors sincerely thank the reviewer for the positive comments and the suggestions on improving the manuscript. Following the reviewer's suggestions, the authors have carefully deleted and summarized the conclusions. All the modifications are printed in red in the revised manuscript. The modifications of the conclusions are as follows:

In this study, taking public participation as an important factor in industrial pollution control into a unified analytical framework, we put forward a two-party game model of government-enterprise and tripartite game model of government-enterprise-public, and explored the multi-agent behavior strategy path of industrial pollution control. According to the economic development situation of China, the government-enterprise and government-enterprise-public game models were assigned values and simulated in scenarios. The main conclusions are as follows:

(1) The government needs to adjust the pollution supervision model. It is not feasible to rely solely on government pollution regulation. It is necessary to add public participation on the basis of a single government led regulatory model, gradually re-form the government regulatory model, and make pollution regulation more efficient. Joining public participation in collaborative regulation is an important breakthrough in solving industrial pollution.

(2) The enterprises strive to reduce the cost of pollution control. Enterprises evade pollution control mainly to reduce production costs and prevent the weakening of their market competitiveness due to increased costs. If enterprises obtain national preferential policies to obtain environmental protection technology and equipment, the cost of pollution control will be significantly reduced, and the tangible and intangible benefits such as economic effects and reputation obtained by enterprises will be higher than the cost of pollution control. The awareness of enterprises and government encouragement policies are of great significance for the development of industrial industries.

(3) There is a positive relationship between the power of public supervision and the active pollution control of enterprises. When enterprises do not actively control pollution and bring negative impacts to the social environment, the supervision power of the public plays a decisive role. The public adopts regulatory strategies to restrain enterprises in pollution control and urge enterprises to achieve positive pollution con-trol effects.

This paper only discusses three stakeholders, i.e. government, the public and enterprises. In fact, there are many more stakeholders involved in industrial pollution. We will establish a multi-participant strategy model to study the influence of other partic-ipants on industrial pollution control in the future.

Reviewer 3 Report

Comments and Suggestions for Authors

The authors have addressed the concerns raised by us, and I am pleased to recommend the manuscript for publication.

Author Response

(The authors gave the same response as above.)

Round 3

Reviewer 2 Report

Comments and Suggestions for Authors

Manuscript ID: sustainability-2803596-R3. Title: Analysis of the dynamic evolution game of government, enterprise and the public to control industrial pollution.

Comments:

1. What was the method used to validate the results of this study? Include in detail in the article. Please indicate the lines where this setting is located.

2. Please contrast your results with other authors. This in the chapter on results and discussion. The discussion of results should be deepened.

Author Response

Response to Comments from Editor and Reviewers

Manuscript ID: sustainability-2803596

Title: Analysis of the dynamic evolution game of government, enterprise and the public to control industrial pollution

Corresponding author: Na Yu *

Listed co-author(s): Meilin Lu

Response to Comments of Reviewer #2

  1. What was the method used to validate the results of this study? Include in detail in the article. Please indicate the lines where this setting is located.

Response: The authors thank the reviewer for the positive comments and the suggestions on improving the manuscript. The method used to validate the results of this study is numerical simulation. The modified section is located from lines 557 to 568

  1. Please contrast your results with other authors. This in the chapter on results and discussion. The discussion of results should be deepened.

Response: The authors appreciated the professional suggestion from the reviewer. Some conclusions of other authors have been added. All the modifications are printed in red in the revised manuscript .
